# RETRIEVAL BACKWARD ATTENTION WITHOUT ADDITIONAL TRAINING: ENHANCE EMBEDDINGS OF LARGE LANGUAGE MODELS VIA REPETITION

## ABSTRACT

The quality of embeddings produced by pretrained language models is critical to downstream performance. Prior work has shown that repeating input text can improve sentence-level representations, but such repetition may adversely affect word-level embeddings. To address this, we propose ReBA (Retrieval Backward Attention), a method combining input repetition with a novel backward attention mechanism that enables tokens to incorporate future context. Across multiple zero-shot tasks, ReBA significantly improves word-level embeddings while preserving sentence-level gains, offering new insights into enhancing representation quality in decoder-only language models. [1]

## 1 INTRODUCTION

Text embedding learning is a key task in Natural Language Processing (NLP) that transforms text into vector representations, making them computationally tractable. Formally, given a text $x$ in a corpus space $X$, embedding learning aims to train a model $f : X \times \Theta \to \mathbb{R}^d$ to generate vector representations $v = f_\theta(x)$ that capture semantic information and enhance performance in downstream tasks such as information retrieval (IR), semantic similarity estimation, classification, and clustering (Ni et al., 2021; Muennighoff et al., 2022).

Advanced large language models (LLMs) have recently demonstrated exceptional generalization and transfer capabilities across downstream tasks. However, Transformer-based models like GPT (Radford, 2018) (unidirectional) and BERT (Kenton & Toutanova, 2019) (bidirectional) are often specialized in different tasks due to the influence of pretraining tasks, with BERT excelling in Natural Language Understanding (NLU) and GPT excelling in Natural Language Generation (NLG). As a result, multiple task-specific models are required (Dong et al., 2019). Research shows that decoder-only models,

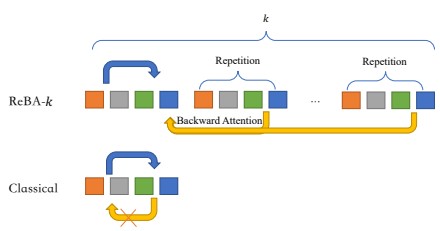

Figure 1: Illustration of ReBA embedding: The classical embedding method captures only the contextual information preceding the token. In contrast, ReBA enhances the quality of target token embeddings by repeating the text $k-1$ times, computing a weighted sum of the target token's original embedding and subsequent token embeddings using backward attention weights. When the sentence appears $k$ times, the resulting embeddings are referred to as ReBA-$k$ embedding.

pre-trained with next-token prediction, achieve superior zero-shot generalization performance across various tasks (Wang et al., 2022), while their performance on NLU is still not good as that of bidirectional models, the performance is highly related to embedding quality, and different approaches to enhance embedding quality have been investigated (Lee et al., 2024; Li et al., 2023; Chen et al., 2024), specifically, some prompting methods have been proposed to improve embedding quality (Jiang et al., 2023; Liu et al., 2024), in this paper, we also focus on improving the embedding quality

---

[1] Our GitHub repository will be made publicly available after the review process. Currently, the code is submitted as anonymized supplementary material.

of decoder-only models from the perspective of the input text by *repetition* and *backward attention* and we call it ReBA embedding (Figure 1).

**Repetition:** We focus on improving the performance of decoder-only models in a zero-shot setting. Research shows that repeating input sentences can provide additional contextual information, significantly boosting model performance. For instance, (Jelassi et al., 2024) found that text repetition allows Transformer models to outperform state-space models like S4 (Gu et al., 2021) and Mamba (Gu & Dao, 2023). Similarly, (Arora et al., 2024; Springer et al., 2025) demonstrate quality improvements in language models through repeated contexts. While repetition boosts decoder-only models, their performance still trails bidirectional models (Table 2, Figure 3). To narrow this gap, we introduce a simple backward attention mechanism that significantly improves context encoding over plain repetition and classical embedding method in zero-shot settings, offering insights for future research.

Figure 2: Illustration of Attention Relationships in BERT and GPT with Corresponding Attention Matrix Representations

**Backward Attention:** In decoder-only architectures, forward attention is typically used, where the attention matrix is a lower triangular matrix. This design lacks associations to subsequent context. Backward attention, on the other hand, represents the relationship between a token and its subsequent context, as reflected in the attention matrix. To enhance the encoding quality of the original text, we propose leveraging repeated text. By concatenating $x$ with itself, represented as $x + x'$, i.e., concatenating $x$ with a copy of itself, the embedding of $x'$ is leveraged to enhance the embeddings of $x$. Since $x'$ appears after $x$, it is natural to consider using backward attention to strengthen $x$'s encoding quality.

Bidirectional models like BERT (Kenton & Toutanova, 2019), RoBERTa (Liu, 2019), XLNet (Yang, 2019), and others (Jiao et al., 2019; Clark, 2020) use symmetric attention matrices (Figure 2). Encoder-decoder models such as T5 (Raffel et al., 2020) permit attention over both past and future tokens, though not in a strictly symmetric manner. However, in such models, the addition of new tokens alters all previous embeddings, thereby increasing computational cost.

In contrast, decoder-only models like GPT (Radford, 2018; Radford et al., 2019; Brown et al., 2020), LLaMA (Touvron et al., 2023a;b), Qwen (Bai et al., 2023), and Baichuan (Baichuan, 2023) use lower triangular attention (Figure 2), where each token attends only to previous ones. This structure avoids recomputation when appending tokens, enabling efficient enhancement of specific token embeddings via backward attention. Our contributions are summarized as follows:

- We propose a novel algorithm that significantly enhances the embedding quality of pretrained models, leading to improved performance in downstream tasks and stronger natural language understanding capabilities.

- The method achieves these improvements without requiring additional training of the model, training new models, or introducing extra parameters, ensuring simplicity and efficiency.

- Our algorithm maintains the advantages of unidirectional LLMs while also capturing subsequent context, enabling targeted enhancement of specific token embeddings.

## 2 PRELIMINARIES

Our goal is to obtain a sentence embedding vector $LLM(C) \in \mathbb{R}^d$ for a sentence $C$ and a word embedding vector $LLM(w|C) \in \mathbb{R}^d$ for word $w$ in context $C$. These vectors serve as semantic representations, and our aim is to improve them to more effectively capture sentence and word-level

semantics. These vectors can measure the similarity between sentences or words, and can be used in downstream tasks such as text classification and word sense disambiguation.

We are particularly interested in extracting these vectors using an *autoregressive language model*. An autoregressive language model predicts the next token in a sequence based on the preceding tokens. This mechanism inherently limits the model to capturing information only from earlier tokens, as reflected in the attention matrix—a lower triangular matrix where each token attends only to its preceding tokens. To enhance the embedding quality, we aim to reutilize the information in the attention matrix to also capture insights from subsequent tokens.

## 2.1 SELF-ATTENTION AS GLOBAL CONTEXT INTEGRATOR

The attention mechanism (Vaswani, 2017) can be understood mathematically as a mapping from a query vector $\mathbf{q}_j \in \mathbb{R}^d$ to a weighted sum of a set of key-value pairs $(\mathbf{k}_i, \mathbf{v}_i)$, where $\mathbf{k}_i, \mathbf{v}_i \in \mathbb{R}^d$, $i$ indexes over the tokens in the input sequence. Formally, given an input sequence of embeddings $\{\mathbf{x}_1, \cdots, \mathbf{x}_n\}$, the attention mechanism computes the output embedding $v_j$ for each token as follows:

$$v_j = \sum_{i=1}^{j} \alpha_{i,j} \mathbf{v}_i, \quad \text{where} \quad \alpha_{i,j} = \frac{\exp(\mathbf{q}_j \cdot \mathbf{k}_i/\sqrt{d})}{\sum_{m=1}^{n} \exp(\mathbf{q}_j \cdot \mathbf{k}_m/\sqrt{d})} \tag{1}$$

Here, $\mathbf{q}_j = \mathbf{W}_q \mathbf{x}_j$, $\mathbf{k}_i = \mathbf{W}_k \mathbf{x}_i$, and $\mathbf{v}_i = \mathbf{W}_v \mathbf{x}_i$ are the query, key, and value vectors, obtained by trainable linear transformations $\mathbf{W}_q, \mathbf{W}_k, \mathbf{W}_v \in \mathbb{R}^{d \times d}$ of the input embeddings, and $d$ is the dimensionality of the query/key space.

In Transformer-based models, each token in the sequence simultaneously acts as a query, key, and value, resulting in contextualized embeddings for all tokens. This mechanism enables the model to integrate information across the entire sequence. For example, given an input sequence $\{w_1, \cdots, w_n\}$, the output embedding $v_n$ for the last token $w_n$ is computed as:

$$v_n = \text{Attention}(\mathbf{q}_n, \{\mathbf{k}_1, \cdots, \mathbf{k}_n\}, \{\mathbf{v}_1, \cdots, \mathbf{v}_n\}), \tag{2}$$

where $\mathbf{q}_n$ is derived from $w_n$ and $\{\mathbf{k}_i, \mathbf{v}_i\}$ are derived from all preceding tokens $\{w_1, \cdots, w_n\}$.

This computation illustrates how attention integrates global context. Notably, in autoregressive models, the attention mechanism is constrained such that $v_i$ only depend on $w_1, \cdots, w_i$, ensuring that information flows unidirectionally. Such a framework blurs the traditional boundary between word and sentence embeddings. The embedding $v_n$ for the final token incorporates contextual information from the entire sequence, making it a natural representation for the full sentence. This marks a fundamental departure from traditional word embedding models like Word2Vec (Mikolov, 2013), which lack explicit mechanisms for modeling token interactions. We will discuss this further in the following section.

## 2.2 WORD EMBEDDING AND SENTENCE EMBEDDING

Traditional models like Word2Vec (Mikolov, 2013) and Glove (Pennington et al., 2014) treat words in isolation, requiring extra processing (e.g., average pooling or Sent2Vec (Moghadasi & Zhuang, 2020)) to construct meaningful sentence embeddings, as individual word vectors lack contextual information.

Unlike traditional word embedding models, Transformer-based LLMs derive each token's embedding through interactions with other tokens. In autoregressive models, given input $w_1, \cdots, w_n$, the output embeddings $v_1, \cdots, v_n$ are generated. While $v_n$ represents $w_n$, it also captures the entire sentence via attention. However, due to unidirectionality, $v_1, \cdots, v_{n-1}$ do not incorporate information from $w_n$. Thus, Transformer-based models blur the traditional distinction between word and sentence embeddings. In particular, the embedding of the last token can serve as an effective representation of the entire sentence.

In this paper, we evaluate the performance of our method on word and sentence embeddings. To assess the quality of word embeddings, we examine the algorithm's performance on word sense disambiguation datasets. To evaluate the quality of sentence embeddings, we consider the algorithm's performance on the C-MTEB.

### 2.3 LANGUAGE MODEL EMBEDDING

We first extract embeddings from the activations of the final hidden layer of the language model. Given a sentence $C = \{w_1, \cdots, w_n\}$, we extract token embeddings $LLM(w_i|C)$, which we regard as word embeddings.

In practice, we consider two main methods for sentence embedding (Wang et al., 2023; Reimers, 2019), one approach called **last token pooling** is to use the embedding of the last token as the sentence embedding: $LLM(w_{-1}|C)$, while the other called **mean token pooling** involves averaging the embeddings of all tokens to obtain the sentence embedding: $\frac{1}{|C|} \sum_{w \in C} LLM(w|C)$. In this paper, we adopt these methods to obtain sentence embeddings.

### 2.4 NEW EMBEDDING VIA REPETITION AND BACKWARD ATTENTION

To enhance the natural language understanding capabilities of autoregressive models while avoiding the costly recomputation of all token embeddings that occurs in bidirectional models when new tokens are added, we propose a novel method called *ReBA* (Retrieval Backward Attention) embedding. This approach leverages the model's inherent capabilities to more effectively capture bidirectional information and enhance model performance.

## 3 MAIN METHOD

The core concept of ReBA is to repeat the input text *twice* and extract the attention matrix from the model. This attention matrix is then applied for backward attention, updating the embeddings of specific tokens based on the repeated text. Finally, the updated and original embeddings are combined to yield the final embedding vectors.

### 3.1 CLASSICAL EMBEDDING IGNORES BIDIRECTIONAL CONTEXT

As discussed in Sec 2.1, classical sentence embeddings fail to effectively capture bidirectional information. In autoregressive language models, the contextualized embedding at position $k$ encodes information only from tokens preceding $k$, without considering subsequent tokens. As a result, tokens at the beginning of a sentence may not fully capture their intended meaning, lacking semantic information from subsequent context.

### 3.2 REPETITION CAPTURES BIDIRECTIONAL CONTEXT

Research shows that text repetition can significantly enhance the bidirectional information captured by sentence embeddings in autoregressive models (Springer et al., 2025; Jelassi et al., 2024). By repeating text, the model's embeddings become more contextually enriched. Rather than using prompts such as "Rewrite the sentence: $x$, rewritten sentence: $x$" as in Springer et al. (2025), we directly repeat the sentence to eliminate prompt effects. For example, given the sentence "I love NLP.", the repeated versions are:

- Repeated once: 'I love NLP. I love NLP.'
- Repeated twice: 'I love NLP. I love NLP. I love NLP.'

For the word 'love', we observe that the second and third occurrences carry more contextual information than the first, as they capture the broader context. Following Springer et al. (2025), we refer to these enhanced embeddings as *Echo embeddings*.

### 3.3 REBA EMBEDDING

Inspired by the benefits of text repetition, we propose a new method that does not require the use of prompts in Springer et al. (2025). Instead of using the embeddings from the repeated text, we introduce backward attention to enhance the original embeddings.

When considering Echo embeddings as word embeddings, they may introduce *information imbalance*. For example, given an original text $C = \{w_1, w_2, w_3\}$ and its repeated version

$C^{new} = \{w_1, w_2, w_3, w_4, w_5, w_6\}$, the Echo approach replaces the embedding of the first $w_2$ with that of the second occurrence $w_5$. However, due to causal masking, $w_5$ cannot access information from the subsequent token $w_6$ and only incorporates context from the preceding $w_4$. This creates an asymmetric and biased representation, potentially degrading the quality of word embeddings.

In contrast, ReBA enhances the embedding of the first $w_2$ by aggregating information from all tokens that follow it in $C^{new}$—namely $w_3, w_4, w_5, w_6$. This ensures contextual symmetry and significantly improves representation quality. Our method proceeds in three steps:

**1) First Step: Construct Attention Matrix Extraction**

Vig & Belinkov (2019) found that the performance of GPT2's multi-head attention matrices varies across different attention heads and hidden layers, and some of the attention heads in the deeper layers contribute less to the model's performance (He et al., 2024). Inspired by this observation, we aim to leverage all attention matrices in a new way to enhance the quality of embedding vectors in the following way:

*We record an interaction between tokens whenever it is captured by any attention head in any layer, and retain the maximum observed attention weight.*

Formally, for any token pair $(i, j)$, the fused attention weight $A_{i,j}^{new}$ is computed as the maximum value observed across all heads and layers:

$$A_{i,j}^{new} = \max_{p,q} \tilde{A}_{i,j}^{p,q}, \quad \text{where} \quad \tilde{A}^{p,q} = \frac{A^{p,q} + (A^{p,q})^T}{2}$$

where $\tilde{A}^{p,q}$ is the symmetrized attention matrix $A^{p,q}$ from the $p$-th head in the $q$-th layer. This approach is grounded in the following rationale:

**Salience Preservation**: If any attention head, in any layer, assigns a high weight to the token pair $(i, j)$, it indicates a strong and potentially meaningful relationship. The max operator ensures that such salient connections are retained.

**Robustness to Redundancy**: Many attention heads may learn redundant or noisy patterns (He et al., 2024). By taking the maximum, we effectively filter out spurious or weak attentions, prioritizing only the most confident signals.

**Interpretability**: The resulting matrix $A^{new}$ highlights the most influential token-wise interactions, providing a clear and interpretable summary of the model's collective attention behavior.

While alternative aggregation schemes (e.g., mean or top-$k$) are possible, the maximum rule offers a simple yet effective mechanism for identifying and preserving the most salient semantic relationships, which is particularly beneficial for downstream embedding enhancement.

---

**Algorithm 1** Attention Matrix Extraction

---

**Input:** Text with length $n$, pretrained language model $LLM$ and its number of hidden layers $I$, number of attention heads $J$.
**Output:** New Attention Matrix $A^{new}$.
1. Extract all attention matrices $A^{p,q} \in \mathbb{R}^{n \times n} \{p = 1, \cdots, I; q = 1, \cdots, J\}$ from the pretrained model, initialize $A^{new} = 0$.
2. **for** $p = 1, 2, \cdots, I$:
$\quad$ **for** $q = 1, 2, \cdots, J$:
$\quad\quad \tilde{A}^{p,q} = \frac{A^{p,q} + (A^{p,q})^T}{2}$
$\quad\quad A^{new} = \frac{A^{new} + \tilde{A}^{p,q}}{2} + \frac{|A^{new} - \tilde{A}^{p,q}|}{2}$
3. **Return** $A^{new}$.

---

**2) Second Step: Backward Attention and Text Repetition**

---

**Algorithm 2** Word Embedding with ReBA Mechanism

---

**Input:** Text sequence $C = \{w_1, w_2, \ldots, w_n\}$, pretrained language model $LLM$, target word $w_i$.
**Output:** Word embedding $e_i$.
1. Duplicate the input: $C^{new} = \{w_1, w_2, \ldots, w_{2n}\}$ as new input.
2. Input the text $C^{new}$ and compute the attention matrix $A^{new}$ using Algorithm 1.
3. Extract the embedding $e_i$ of $w_i$ $\{i \leq n\}$ based on $A^{new}$: $e_i = \sum_{k=i}^{2n} \alpha'_{i,k} v_k$ where $\alpha'_{i,k} = A^{new}_{i,k}$ is the $i$-th value of $k$-th column of $A^{new}$ and $v_k = LLM(w_k|C)$.
5. **Return** $e_i$.

---

---

**Algorithm 3** Sentence Embedding with ReBA Mechanism

---

**Input:** Text sequence $C = \{w_1, w_2, \ldots, w_n\}$, pretrained language model $LLM$.
**Output:** Sentence embedding $E$.
1. Extract word embeddings $e_i$ for each token $w_i$ in $C$ using Algorithm 2.
2. **If** last token pool: $E = e_n$.
**Else if** mean token pool: $E = \frac{1}{n}\sum_{i=1}^{n} e_i$.
3. **Return** $E$.

---

To compute text embeddings effectively, we integrate *text repetition* and a *backward attention* mechanism as outlined in Algorithm 2 (for word embedding) and Algorithm 3 (for sentence embedding). Given a text sequence $C = \{w_1, w_2, \ldots, w_n\}$, we first apply *text repetition* by duplicating the sequence to form a new input:

$$C^{new} = \{w_1, w_2, \ldots, w_n, w_{n+1}, \ldots, w_{2n}\}$$

where $w_i$ with $i > n$ represents the repeated token.

Next, we compute the attention matrix $A^{new} \in \mathbb{R}^{2n \times 2n}$ with Algorithm 1 to capture the contextual dependencies across both the original and repeated sequences.

The backward attention mechanism is applied to strengthen semantic propagation by iteratively tracing the connections from the repeated tokens $\{w_{n+1}, w_{n+2}, \ldots, w_{2n}\}$ back to the original sequence $\{w_1, w_2, \ldots, w_n\}$. For a target token $w_i \in C$, the embedding $e_i$ is computed as:

$$e_i = \sum_{k=i}^{2n} \alpha'_{i,k} v_k, \text{(backward attention)} \tag{3}$$

where $\alpha'_{i,k} = A^{new}_{i,k}$ is the attention weight from token $w_k$ to $w_i$, and $v_k = LLM(w_k, C^{new})$ represents the contextualized embedding of the token $w_k$.

This approach propagates semantic relationships—especially from later tokens—into the final representation.

**3) Final Step: Embedding Vector Construction**

Under different pooling strategies, the resulting sentence vectors vary. For the case of **last token pooling**, we directly use $e_n$ as the sentence encoding. In contrast, for **mean token pooling**, we take the average of all $e_i$, defined as $\frac{1}{n}\sum_{i=1}^{n} e_i$, as the sentence embedding. Notably, the order of summation can be exchanged for efficient computation:

$$\frac{1}{n}\sum_{i=1}^{n} e_i = \frac{1}{n}\sum_{i=1}^{n}\sum_{k=i}^{2n} \alpha'_{i,k} v_k = \frac{1}{n}\sum_{k=1}^{2n}\sum_{i=1}^{\min(n,k)} \alpha'_{i,k} v_k = \frac{1}{n}\sum_{k=1}^{2n} \alpha'_k v_k,$$

where $\alpha'_k = \sum_{i=1}^{\min(n,k)} \alpha'_{i,k}$ is the sum of the $k$-th column of $A^{new}$. By exchanging the order of summation, we can first sum over each column of $A^{new}$ and then perform the remaining calculations. This reduces the original computational complexity from $O(n^2)$ to $O(n)$.

For word embeddings, we compute the embedding $e_i$ of $w_i$ directly using Eq. 3, without requiring additional operations. The detailed computation process of all methods we use is shown in Table 1.

Table 1: Introduction of our experimental settings. (1) In both ReBA and Echo method we need to repeat the original sentence, We use $w_1, \cdots, w_n$ to denote the original input tokens, and the repeated tokens are $w_k = w_{k\%n}$ when $k > n$, $e_i$ is the new embeddings. (2) $v_i$ is the original output of $w_i$ which is denoted as $v_i := LLM(w_i|C)$ where $C$ is the context , and $\alpha'_{i,j}$ is the $i$-th value of $j$-th column of the attention weight extracted by Algorithm 1. (3) ReBA-1 is equivalent to classical sentence embedding evaluation with last pooling strategy so we only test ReBA-1 for word embedding evaluation.

| Method | Input Tokens | Embedding for Evaluation | | |
|---|---|---|---|---|
| | | Word Embedding for $w_i$ | Sentence Embedding (Mean Pooling) | Sentence Embedding (Last Pooling) |
| ReBA-$k$ (Ours) | $\{w_1, \cdots, w_{kn}\}$ | $e_i = \sum_{j=i}^{kn} \alpha'_{i,j} v_j$ | $\frac{1}{n} \sum_{j=1}^{n} e_j$ | $e_n$ |
| Echo-$k$ | $\{w_1, \cdots, w_{kn}\}$ | $v_{(k-1)n+i}$ | $\frac{1}{(k-1)n} \sum_{j=n}^{kn} v_j$ | $v_{kn}$ |
| Classical | $\{w_1, \cdots, w_n\}$ | $v_i$ | $\frac{1}{n} \sum_{j=1}^{n} v_j$ | $v_n$ |

Table 2: *Sentence Embeddings*: Zero-shot average scores on C-MTEB to evaluate the performance of *Sentence Embeddings*. The top two rows are the main results, testing the scores of ReBA, simple repetition and traditional encoding on GPT-2-Chinese and LLaMA-2-Chinese-7B. The third row is an ablation experiment, testing different pooling strategies. The fourth row is an ablation experiment with more repetitions, testing the effect of repeating twice. The fifth row is a comparison experiment with bidirectional model using mean pooling or the first token 'CLS', BERT refers to bert-base-chinese model with 102M params here.

| Strategy | Model | Pool | Clas. | P. Cls. | Clus. | Retr. | STS | Rera. | Total Average |
|---|---|---|---|---|---|---|---|---|---|
| Main results: | | | | | | | | | |
| ReBA-2 (Ours) | GPT-2 | Last | **0.5414** | 0.5422 | **0.3243** | **0.2173** | **0.2118** | **0.2979** | **0.3634** |
| Echo-2 | GPT-2 | Last | 0.5010 | **0.5456** | 0.2255 | 0.1757 | 0.1756 | 0.2460 | 0.2907 |
| Classical | GPT-2 | Last | 0.4990 | 0.5413 | 0.2305 | 0.1474 | 0.1446 | 0.2101 | 0.2590 |
| ReBA-2 (Ours) | LLaMA-2 | Last | **0.6889** | 0.5225 | 0.3918 | **0.6640** | 0.1949 | 0.3653 | **0.4801** |
| Echo-2 | LLaMA-2 | Last | 0.6759 | **0.5503** | **0.4136** | 0.5629 | **0.2636** | **0.3784** | 0.4733 |
| Classical | LLaMA-2 | Last | 0.6574 | 0.5228 | 0.3354 | 0.4667 | 0.1506 | 0.2816 | 0.3951 |
| Ablations: Different Pool Strategy | | | | | | | | | |
| ReBA-2 (Ours) | GPT-2 | Mean | **0.5538** | 0.5554 | **0.3275** | **0.4484** | **0.2567** | **0.3000** | **0.3977** |
| Echo-2 | GPT-2 | Mean | 0.5480 | **0.5588** | **0.3275** | 0.3691 | 0.2360 | 0.2964 | 0.3734 |
| Classical | GPT-2 | Mean | 0.5465 | 0.5452 | 0.3166 | 0.2959 | 0.2515 | 0.2959 | 0.3499 |
| Ablations: More Repetitions | | | | | | | | | |
| Echo-3 | GPT-2 | Last | 0.4973 | **0.5473** | 0.1969 | 0.1845 | 0.1845 | 0.2488 | 0.2924 |
| ReBA-3 (Ours) | GPT-2 | Last | **0.5422** | 0.5446 | **0.3246** | **0.2119** | **0.2284** | **0.3010** | **0.3699** |
| Comparison: Bidirectional model | | | | | | | | | |
| Classical | BERT | Mean | 0.6734 | 0.5435 | 0.4225 | 0.5537 | 0.2594 | 0.3445 | 0.4658 |
| Classical | BERT | CLS | 0.6628 | 0.5604 | 0.3598 | 0.3003 | 0.2070 | 0.2546 | 0.3679 |

## 3.4 TIME EFFICIENCY OF REBA

Assuming the model has $I$ hidden layers, each with $J$ attention heads, and an input sequence of length $n$, the traditional method has a time complexity of $O(IJn^2)$. This arises from extracting $I \times J$ attention matrices, each of size $n \times n$, and performing operations like symmetry computation and maximum attention update, both of which take $O(n^2)$ for a single matrix.

# 4 EXPERIMENTS

## 4.1 DATASETS

We will test our method on several datasets, including text classification, text clustering, text pair classification, text reranking, text retrieval, sentence similarity, etc. In particular, we will also test its performance on Chinese polysemous word understanding task.

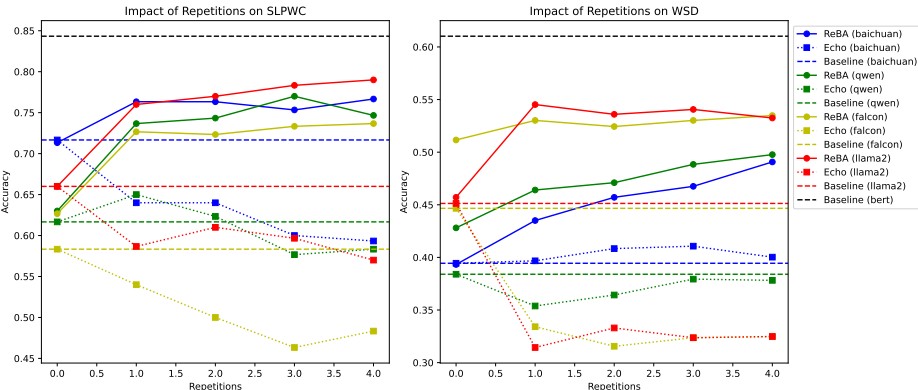

Figure 3: *Word Embeddings*: Performance on SLPWC and WSD tasks using Euclidean distances to evaluate *Word Embeddings*. The results show that ReBA encoding significantly enhances model performance on polysemous word understanding tasks. While performance fluctuates with the number of repetitions, increasing the repetition count does not necessarily lead to significant improvements. Based on this experiment, we observe that simple sentence repetition is not effective for improving word-level embeddings and only contributes to sentence-level understanding. Furthermore, the backward attention mechanism remains crucial for achieving further performance enhancements.

**1) Chinese Massive Text Embedding Benchmark (For Sentence Embedding)**

To evaluate sentence embedding, we use the Chinese Massive Text Embedding Benchmark (C-MTEB). It is a collection of datasets from six categories: classification, clustering, pair classification, reranking, retrieval, sentence similarity, containing 31 datasets in total. Due to the scale of this dataset, we will only test two unidirectional language models: the fine-tuned versions of GPT-2 and LLaMA-2-7B's Chinese pre-trained models.

**2) Chinese Polysemous Word Disambiguation Dataset (For Word Embedding)**

These datasets evaluate a model's ability to understand polysemous words and assess word embedding performance. The first dataset, the Sentence Level Polysemous Words Classification (SLPWC) subset of the Chinese Semantic Evaluation Dataset [2] (C-SEM) benchmark, contains 300 questions where the task is to identify which option represents a different meaning of a polysemous word. We extract embeddings of the target word in different contexts using an LLM, and the correct answer is determined by summing the embedding distances (Cosine or Euclidean). The second dataset, Word Sense Disambiguation (WSD, (Yan et al., 2023)), is a Chinese semantic dataset used for word sense disambiguation tasks. We evaluate performance based on accuracy and adapt it into a 4-choice format to suit our algorithm.

## 4.2 RESULTS

In our experiments, we use pretrained autoregressive models based on the Transformers architecture. These include fine-tuned versions of GPT-2 and LLaMA-2's Chinese pretrained models (GPT-2-Chinese, LLaMA-2-Chinese-7B), as well as models like Qwen-7B, BaiChuan-7B, and Falcon-7B, which are suitable for evaluation on Chinese datasets. All model details are presented in Appendix A.3.

The overall results (see Table 2 and Figure 3) show that the ReBA encoding method significantly outperforms traditional encoding methods for both sentence and word embeddings. In particular, our method demonstrates a significant improvement in accuracy on semantic understanding tasks. This indicates that our algorithm effectively enhances language models' understanding of natural language in zero-shot setting.

---

[2]Accessible at: `https://github.com/FlagOpen/FlagEval/tree/master/csem`

### 4.2.1 SENTENCE EMBEDDING EVALUATION

We primarily evaluate GPT-2-Chinese (we call it GPT-2 for simplicity) and LLaMA-2-Chinese-7B (LLaMA-2, for simplicity) on C-MTEB, and compare the performance of our method with traditional encoding approaches under different pooling strategies. Across nearly all tasks, our method demonstrates significant improvements over traditional methods. Results are concluded as follows:

(1) *ReBA consistently enhances unidirectional language models.* ReBA achieves greater improvements on GPT-2 compared to simple repetition. For LLaMA-2, while our method outperforms traditional encoding, the gains on certain tasks (e.g., STS) are less pronounced compared to simple repetition;

(2) *Increasing the repetition count does not yield additional benefits.* We compare the effects of repeating once and twice (ReBA-2 and ReBA-3), finding that while repeating twice achieves better results, the improvement is marginal. This conclusion holds for both ReBA and simple repetition;

(3) *Our method remains effective across different pooling strategies.* With last-pooling, the algorithm achieves substantial improvements, while the gains with mean-pooling are comparatively smaller.

### 4.2.2 WORD EMBEDDING EVALUATION

We evaluate the performance of LLaMA-2-Chinese-7B, Qwen-7B, BaiChuan-7B, and Chinese-Falcon-7B and Bert-base-chinese (102M) on the Chinese polysemous word understanding task using the Chinese SEMantic evaluation dataset (C-SEM) and the Word Sense Disambiguation dataset (WSD). Results are presented in Figure 3:

(1) *ReBA encoding performs exceptionally well as a word embedding method.* ReBA significantly surpasses classical embeddings;

(2) *Backward attention is the essential operation.* We test Echo embeddings as word embeddings by using the embedding of the target word's last occurrence. Surprisingly, repetition alone degrades performance, but adding backward attention significantly improves it. ReBA embeddings outperform both Echo and classical embeddings;

(3) *Increasing the repetition count does not yield additional benefits.* Also, in the word embedding task, we find that repeating twice does not bring more benefits than repeating once. This conclusion is consistent with the sentence embedding task.

## 5 CONCLUSION AND DISCUSSION

We propose a context-enhanced encoding method using backward attention and repetition, achieving notable improvements on LLMs across tasks. On sentence embedding evaluation datasets, we observe that the backward attention mechanism was not a decisive factor—simply repeating sentences was sufficient to improve sentence vector quality. This effect was particularly pronounced in larger models like LLaMA-2, where the gains from backward attention are minimal.

However, on word embedding evaluation datasets, the backward attention mechanism plays a crucial role. In these cases, simply repeating sentences leads to performance degradation, whereas incorporating backward attention results in substantial improvements. Consequently, ReBA is a more general method for improving language model encoding quality.

Now we discuss a possible improvement measure, repeating the entire text doubles the sequence length, introducing an additional computational overhead of $O(L^2)$. We propose a potential alternative for future study: First encode the original sequence $S$ of length $L$ using an LLM to obtain the initial embeddings. Next, we divide $S$ into subsequences $S_0, S_1, \ldots, S_n$, each of length $L_0$. For each subsequence $S_i$, we repeat it to form $2S_i$ and then extract the attention weights between the original and repeated embeddings of $S_i$, we apply backward attention to enhance the initial embeddings of $S_i$, then we can extract the embeddings of word appears in $S_i$ for word embedding or use mean pooling to get the whole sentence embedding. This approach reduces the additional computational overhead caused by repetition from $O(L^2)$ to $O(L \cdot L_0)$. When $L_0$ is small enough, the additional cost becomes negligible.

## ETHICS STATEMENT

This work does not involve human subjects, personally identifiable information, or other sensitive content. All datasets used are publicly available and used in accordance with their licenses. Our method does not introduce any foreseeable ethical risks, harm, or bias beyond standard use of pretrained language models.

## REPRODUCIBILITY STATEMENT

In this work, we introduce ReBA, a method for enhancing word and sentence embeddings in autoregressive language models via text repetition and backward attention. All details necessary to reproduce the reported results are provided in the main text and appendix. Specifically, Algorithm 1–3 describe the embedding extraction and backward attention procedure. Model configurations, hyperparameters, and data preprocessing steps for GPT-2-Chinese, LLaMA-2-Chinese-7B, and other evaluated models are provided in Appendix A.3. Experiments on C-MTEB and Chinese polysemous word disambiguation datasets follow standard evaluation protocols. We have no random initialization, and all experiments use publicly available datasets and pretrained models. Our code is currently submitted as supplementary material. We plan to post the official GitHub link in the main text after publication.

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

# A APPENDIX

## A.1 THE USE OF LARGE LANGUAGE MODELS (LLMS)

In this work, we employed a large language model (GPT-4) exclusively for language polishing and proofreading. The model assisted in improving the fluency and clarity of certain passages, particularly in the introduction and discussion sections. All scientific contributions, including core ideas, methodologies, experiments, and conclusions, originate entirely from the authors. The LLM was used solely as a writing aid, with all outputs carefully verified to ensure technical accuracy.

## A.2 DATASETS

### A.2.1 SENTENCE EMBEDDING EVALUATION

To evaluate sentence embeddings, we use the Chinese Massive Text Embedding Benchmark (C-MTEB), which is a collection of datasets across six categories: classification, clustering, pair classification, reranking, retrieval, and sentence similarity. In total, there are 31 datasets. The dataset is available at https://huggingface.co/datasets/C-MTEB, and the leaderboard can be found at https://huggingface.co/spaces/mteb/leaderboard.

**Data preprocessing:** A brief overview of the C-MTEB datasets is provided below:

In our experiments, since GPT models have an input sequence limit of 512 tokens, we applied text truncation accordingly. By analyzing the text length distribution in the C-MTEB dataset, we found that most texts are under 512 tokens (Figure 4), and a substantial portion remains below 1024 tokens even when repeated three times. Therefore, for our experiments with LLaMA-2, in order to avoid potential memory overflow issues caused by a small number of long texts, we truncated the text to a maximum length of 1024 tokens.

The detailed results can be found in Table 2. Additionally, Table 3 presents the results for GPT-2 on C-MTEB using the last pooling strategies.

### A.2.2 WORD EMBEDDING EVALUATION

To evaluate word embeddings, we use the Chinese SEMantic evaluation dataset (C-SEM), a benchmark dataset for semantic evaluation. We use the Sentence Level Polysemous Words Classification

| Task Type | Task Name | Classical | Echo-2 | Echo-3 | ReBA | ReBA-3 |
|---|---|---|---|---|---|---|
| Classification | TNews | 0.2775 | 0.2679 | 0.2766 | 0.3061 | 0.3055 |
| Classification | IFlyTek | 0.2288 | 0.1915 | 0.2099 | 0.3094 | 0.3099 |
| Classification | MultilingualSentiment | 0.4851 | 0.4838 | 0.4865 | 0.5044 | 0.5064 |
| Classification | JDReview | 0.6989 | 0.7092 | 0.7069 | 0.7357 | 0.7342 |
| Classification | OnlineShopping | 0.6637 | 0.6626 | 0.6681 | 0.6943 | 0.6948 |
| Classification | Waimai | 0.6398 | 0.6687 | 0.6580 | 0.6991 | 0.7026 |
| Clustering | CLSClusteringS2S | 0.1282 | 0.1505 | 0.1446 | 0.2289 | 0.2308 |
| Clustering | CLSClusteringP2P | 0.2086 | 0.1077 | 0.1480 | 0.3081 | 0.3107 |
| Clustering | ThuNewsClusteringS2S | 0.2419 | 0.2587 | 0.2678 | 0.3250 | 0.3246 |
| Clustering | ThuNewsClusteringP2P | 0.3433 | 0.2708 | 0.3417 | 0.4352 | 0.4322 |
| Pair Classification | Ocnli | 0.5338 | 0.5306 | 0.5349 | 0.5382 | 0.5382 |
| Pair Classification | Cmnli | 0.5488 | 0.5639 | 0.5562 | 0.5462 | 0.5511 |
| Reranking | T2Reranking | 0.5254 | 0.5539 | 0.5462 | 0.5570 | 0.5613 |
| Reranking | MMarcoReranking | 0.0263 | 0.0574 | 0.0535 | 0.0709 | 0.0714 |
| Reranking | CMedQAv1 | 0.1399 | 0.1800 | 0.1852 | 0.2788 | 0.2811 |
| Reranking | CMedQAv2 | 0.1488 | 0.2037 | 0.1993 | 0.2852 | 0.2903 |
| Retrieval | T2Retrieval | 0.0564 | 0.1018 | 0.0972 | 0.2079 | 0.2265 |
| Retrieval | MMarcoRetrieval | 0.2201 | 0.3305 | 0.3316 | 0.4766 | 0.4925 |
| Retrieval | DuRetrieval | 0.1076 | 0.1456 | 0.1512 | 0.3076 | 0.3417 |
| Retrieval | CovidRetrieval | 0.0464 | 0.0105 | 0.0200 | 0.1628 | 0.1786 |
| Retrieval | CmedqaRetrieval | 0.2199 | 0.3171 | 0.3105 | 0.4196 | 0.4188 |
| Retrieval | EcomRetrieval | 0.2430 | 0.3880 | 0.3660 | 0.5330 | 0.5620 |
| Retrieval | MedicalRetrieval | 0.0820 | 0.1700 | 0.1520 | 0.2860 | 0.2990 |
| Retrieval | VideoRetrieval | 0.2040 | 0.4490 | 0.3710 | 0.5670 | 0.5810 |
| STS | ATEC | 0.1321 | 0.1476 | 0.1453 | 0.1652 | 0.1769 |
| STS | BQ | 0.1902 | 0.2239 | 0.2134 | 0.2532 | 0.2561 |
| STS | LCQMC | 0.1116 | 0.2111 | 0.1830 | 0.3199 | 0.3329 |
| STS | PAWSX | 0.1234 | 0.1185 | 0.1195 | 0.1331 | 0.1327 |
| STS | STSB | 0.2563 | 0.3429 | 0.3268 | 0.3469 | 0.3660 |
| STS | AFQMC | 0.0798 | 0.0780 | 0.0789 | 0.0997 | 0.1060 |
| STS | QBQTC | 0.1189 | 0.1696 | 0.1628 | 0.1651 | 0.1512 |
| Total average | N/A | 0.2591 | 0.2924 | 0.2907 | 0.3634 | 0.3699 |

Table 3: Main Results: Zero-shot scores of GPT-2 models on C-MTEB under last pooling strategy with different methods. 'Classical' refers to the traditional encoding method, 'Echo-2' and 'Echo-3' refer to the methods that only repeat the text without backward attention, and 'ReBA' and 'ReBA-3' refer to our proposed method with one and two repetitions, respectively, the scores we choose are 'accuracy', 'v_measure' , 'map', 'cos_sim :accuracy', 'cos_sim :pearson', 'recall_at_1000' for Classification, Clustering, Reranking, Pair Classification, STS, Retrieval.

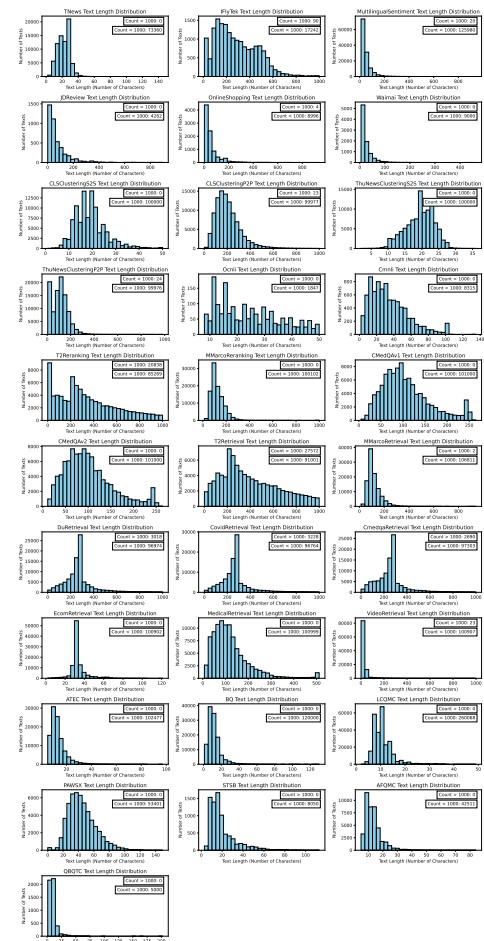

Figure 4: Information about C-MTEB, with most text lengths within 1000 tokens.

(SLPWC) subset of C-SEM as our evaluation dataset. This subset is designed to test a model's ability to understand polysemy (i.e., words with multiple meanings). The evaluation involves presenting a word in different contexts and expecting the model to identify semantic differences.

1. **SLPWC**:

The SLPWC dataset contains 300 polysemous words, each of which appears in four sentences. In three of the sentences, the polysemous word has the same meaning, while in the remaining sentence, the word has a different meaning. The task is to identify the sentence with the different meaning, the data presents a question format: 'Which of the following sentences uses 'word' differently from the others? A. sentence1; B. sentence2; C. sentence3; D. sentence4.' An example from the dataset is presented in section A.2.2.

2. **WSD**:

The Word Sense Disambiguation (WSD) dataset contains 1,023 polysemous words, each associated with multiple meanings, and each meaning linked to several example sentences. The dataset is structured as: {word: {sense1: [sentence1, sentence2, sentence3]; sense2: [sentence4]}}. To ensure consistent evaluation, we converted the WSD dataset into the SLPWC format. Specifically, three sentences are randomly selected from one meaning and one from another. The transformed data presents a question format: 'Which of the following sentences uses 'word' differently from the others? A. sentence1; B. sentence2; C. sentence3; D. sentence4.' Below is an example of question in these two dataset:

| Task Type | Task Name | Classical | Echo-2 | ReBA-2 |
|---|---|---|---|---|
| Classification | TNews | 0.3048 | 0.3090 | 0.3102 |
| Classification | IFlyTek | 0.3414 | 0.3179 | 0.3474 |
| Classification | MultilingualSentiment | 0.4952 | 0.5062 | 0.5058 |
| Classification | JDReview | 0.7272 | 0.7432 | 0.7373 |
| Classification | OnlineShopping | 0.6957 | 0.7021 | 0.7028 |
| Classification | Waimai | 0.7150 | 0.7095 | 0.7193 |
| Clustering | CLSClusteringS2S | 0.2188 | 0.2473 | 0.2430 |
| Clustering | CLSClusteringP2P | 0.3186 | 0.3189 | 0.3208 |
| Clustering | ThuNewsClusteringS2S | 0.3065 | 0.3222 | 0.3217 |
| Clustering | ThuNewsClusteringP2P | 0.4225 | 0.4214 | 0.4245 |
| Pair Classification | Ocnli | 0.5452 | 0.5501 | 0.5485 |
| Pair Classification | Cmnli | 0.5452 | 0.5675 | 0.5624 |
| Reranking | T2Reranking | 0.5553 | 0.5518 | 0.6061 |
| Reranking | MMarcoReranking | 0.0365 | 0.0629 | 0.0431 |
| Reranking | CMedQAv1 | 0.2326 | 0.2771 | 0.2680 |
| Reranking | CMedQAv2 | 0.2587 | 0.2941 | 0.2833 |
| Retrieval | T2Retrieval | 0.1360 | 0.1992 | 0.3603 |
| Retrieval | MMarcoRetrieval | 0.2843 | 0.4798 | 0.4190 |
| Retrieval | DuRetrieval | 0.1698 | 0.2856 | 0.3261 |
| Retrieval | CovidRetrieval | 0.3335 | 0.1581 | 0.7758 |
| Retrieval | CmedqaRetrieval | 0.3590 | 0.4296 | 0.3962 |
| Retrieval | EcomRetrieval | 0.3860 | 0.5550 | 0.4970 |
| Retrieval | MedicalRetrieval | 0.1660 | 0.2680 | 0.2360 |
| Retrieval | VideoRetrieval | 0.5330 | 0.5780 | 0.5770 |
| STS | ATEC | 0.1678 | 0.1950 | 0.1886 |
| STS | BQ | 0.2918 | 0.2761 | 0.2926 |
| STS | LCQMC | 0.4296 | 0.3853 | 0.4352 |
| STS | PAWSX | 0.1448 | 0.1036 | 0.1214 |
| STS | STSB | 0.4450 | 0.5140 | 0.4917 |
| STS | AFQMC | 0.1018 | 0.1117 | 0.1105 |
| STS | QBQTC | 0.1799 | 0.1366 | 0.1571 |
| Total average | N/A | 0.3499 | 0.3734 | 0.3977 |

Table 4: Ablations: Zero-shot scores of GPT-2 models on C-MTEB under mean pooling strategy with different methods. 'Classical' refers to the traditional encoding method, 'Echo-2' refer to the methods that only repeat the text without backward attention, and 'ReBA' refer to our proposed method with one repetition, respectively, the scores we choose are 'accuracy', 'v_measure' , 'map', 'cos_sim', 'pearson', 'recall_at_1000'.

### A.2.3 ATTENTION MATRIX PROCESSING

We also conducted comparative experiments with symmetric attention matrices and last-layer-only attention. The results are shown in Table 2. Both methods underperformed compared to our approach.

| Task Type | Task Name | Classical | Echo-2 | Echo-3 | ReBA-2 |
|---|---|---|---|---|---|
| Classification | TNews | 0.5194 | 0.5264 | 0.5203 | 0.5334 |
| Classification | IFlyTek | 0.3663 | 0.4082 | 0.3552 | 0.4429 |
| Classification | MultilingualSentiment | 0.6474 | 0.6554 | 0.6476 | 0.6645 |
| Classification | JDReview | 0.7645 | 0.7717 | 0.7497 | 0.7976 |
| Classification | OnlineShopping | 0.8521 | 0.8724 | 0.8674 | 0.8745 |
| Classification | Waimai | 0.7951 | 0.8214 | 0.8226 | 0.8206 |
| Clustering | CLSClusteringS2S | 0.2446 | 0.2836 | 0.3000 | 0.3021 |
| Clustering | CLSClusteringP2P | 0.2858 | 0.3169 | 0.3049 | 0.2972 |
| Clustering | ThuNewsClusteringS2S | 0.4841 | 0.5572 | 0.5522 | 0.5702 |
| Clustering | ThuNewsClusteringP2P | 0.3272 | 0.4968 | 0.5216 | 0.3976 |
| Pair Classification | Ocnli | 0.5181 | 0.5463 | 0.5355 | 0.5176 |
| Pair Classification | Cmnli | 0.5276 | 0.5543 | 0.5498 | 0.5275 |
| Reranking | T2Reranking | 0.5949 | 0.5806 | 0.5708 | 0.6185 |
| Reranking | MMarcoReranking | 0.0437 | 0.0774 | 0.0867 | 0.0619 |
| Reranking | CMedQAv1 | 0.2390 | 0.3976 | 0.3948 | 0.3667 |
| Reranking | CMedQAv2 | 0.2489 | 0.4581 | 0.4713 | 0.4142 |
| Retrieval | T2Retrieval | 0.4387 | 0.4699 | 0.3840 | 0.6334 |
| Retrieval | MMarcoRetrieval | 0.6340 | 0.7425 | 0.7423 | 0.7923 |
| Retrieval | DuRetrieval | 0.5701 | 0.7322 | 0.6740 | 0.7936 |
| Retrieval | CovidRetrieval | 0.5896 | 0.2819 | 0.1923 | 0.6723 |
| Retrieval | CmedqaRetrieval | 0.3713 | 0.6642 | 0.6775 | 0.5355 |
| Retrieval | EcomRetrieval | 0.5990 | 0.7180 | 0.7700 | 0.8310 |
| Retrieval | MedicalRetrieval | 0.2090 | 0.4760 | 0.5030 | 0.4170 |
| Retrieval | VideoRetrieval | 0.3220 | 0.4190 | 0.5780 | 0.6370 |
| STS | ATEC | 0.1328 | 0.1880 | 0.1793 | 0.1876 |
| STS | BQ | 0.1852 | 0.3185 | 0.3199 | 0.2385 |
| STS | LCQMC | 0.2397 | 0.4924 | 0.4886 | 0.3136 |
| STS | PAWSX | 0.1113 | 0.1402 | 0.1399 | 0.1112 |
| STS | STSB | 0.2656 | 0.4559 | 0.4187 | 0.3443 |
| STS | AFQMC | 0.1095 | 0.1549 | 0.1373 | 0.1488 |
| STS | QBQTC | 0.0105 | 0.0956 | 0.1907 | 0.0206 |
| Total average | N/A | 0.3951 | 0.4733 | 0.4724 | 0.4801 |

Table 5: Ablations: Zero-shot scores of LLaMA-2 model on C-MTEB under last pooling strategy with different methods. 'Classical' refers to the traditional encoding method, 'Echo-2' refer to the methods that only repeat the text without backward attention, and 'ReBA-2' refer to our proposed method with one repetition, respectively, the scores we choose are 'accuracy', 'v_measure', 'map', 'cos_sim', 'pearson', 'recall_at_1000'.

## A.3 MODEL DETAIL

Here we provide some details of the models we use in our experiments: BERT-base-chinese[3], GPT2-base-chinese[4], Chinese-llama-2-7b[5], Qwen-7B[6], Chinese-Falcon-7b[7] and BaiChuan-7B[8] as our models.

The BERT-base-chinese model, developed by Google, is a pre-trained language model tailored for Chinese natural language processing tasks. Built on the BERT architecture, it comprises 12 layers, 768

---

[3]BERT-base-chinese:https://huggingface.co/google-bert/bert-base-chinese

[4]GPT2-base-chinese:https://huggingface.co/ckiplab/gpt2-base-chinese

[5]Chinese-llama-2-7b:https://huggingface.co/LinkSoul/Chinese-Llama-2-7b

[6]Qwen-7B:https://huggingface.co/Qwen/Qwen-7B

[7]Falcon-7B: https://huggingface.co/Linly-AI/Chinese-Falcon-7B

[8]BaiChuan-7B:https://huggingface.co/baichuan-inc/Baichuan-7B

hidden units, and 12 attention heads, totaling approximately 110 million parameters. Pre-trained on large Chinese corpora, including Chinese Wikipedia, using Masked Language Modeling (MLM) and Next Sentence Prediction (NSP) objectives, it effectively captures word and sentence-level semantics. This model serves as a robust baseline for tasks such as text classification, named entity recognition, and question answering, offering strong performance across diverse Chinese NLP applications.

The GPT series is a family of pretrained models based on the Transformer architecture, with GPT-2 (Radford et al., 2019) being the second-generation generative pretrained model released by OpenAI in 2019. We used the Chinese version of GPT-2, GPT2-base-chinese, which is fine-tuned on Traditional Chinese datasets to better adapt to Chinese contexts. It has 12 layers, 768 hidden units, and 12 attention heads.

LLaMA-2 (Large Language Model Meta AI 2) (Touvron et al., 2023a) is the second-generation LLM released by Meta (formerly Facebook), designed to handle various language tasks, including text generation, comprehension, and question-answering. It is an enhanced version of the original LLaMA model, featuring improved performance and adaptability. We used a Simplified Chinese fine-tuned version of LLaMA-2 for our experiments. It has 32 layers, 4096 hidden units, and 32 attention heads.

Qwen-7B (Tongyi Qianwen) is a unidirectional language model developed by Alibaba Group. With 7 billion parameters, it is designed to handle a wide range of tasks, including text generation, content summarization, and intelligent decision-making. The model excels in Chinese language processing and supports multilingual tasks, making it suitable for diverse real-world applications.

Baichuan-7B is a unidirectional language model with 7 billion parameters, developed in China for Chinese and multilingual NLP tasks. It demonstrates strong capabilities in machine translation, text classification, and semantic understanding. The model is widely recognized for its adaptability and practical application across various industries.

The Chinese-Falcon-7B, developed by Linly-AI, is an adaptation of the original Falcon architecture, tailored specifically for Chinese natural language processing tasks. With 32 Transformer layers, 71 attention heads per layer, and a hidden size of 4544, it retains the efficient design of Falcon while being pre-trained on a large-scale Chinese corpus. This specialization enables superior performance in Chinese text understanding and generation, making it suitable for applications such as summarization, sentiment analysis, and conversational AI.

Here are the basic statistics of the models used in our experiments:

| Model | Layers | Hidden Units | Heads |
|---|---|---|---|
| BERT | 12 | 768 | 12 |
| GPT-2 | 12 | 768 | 12 |
| LLaMA-2 | 32 | 4096 | 32 |
| Qwen | 32 | 4096 | 32 |
| BaiChuan | 32 | 4096 | 32 |
| Falcon | 32 | 4544 | 71 |

Table 6: Basic Statistics of the Models

## A.4 Details of the Sentence Evaluation: Task Description and Metrics

There are six types of tasks in the C-MTEB dataset: Classification, Clustering, Pair Classification, Reranking, Retrieval, and STS. Each task has specific evaluation metrics and requirements, as detailed below:

The **Classification** task involves assigning labels to text inputs from predefined categories. For example, the TNews dataset requires predicting news categories based on headlines. The primary evaluation metric for this task is accuracy, defined as:

$$\text{Accuracy} = \frac{N_c}{N}$$

where $N_c$ is the number of correct predictions, and $N$ is the total number of samples.

The **Clustering** task groups text samples based on their semantic similarity without predefined labels. An example is the CLSClusteringS2S dataset, where similar sentences need to be grouped together. The evaluation metric is V-measure, defined as:

$$V = 2 \times \frac{H \times C}{H + C}$$

where $H$ represents homogeneity, and $C$ represents completeness.

The **Reranking** task focuses on reordering retrieved documents by their relevance to a query. For instance, in the T2Reranking dataset, the task involves ranking candidate documents for search queries. The main evaluation metric is Mean Average Precision (MAP). For a query $q$, the Average Precision (AP) is defined as:

$$AP_q = \frac{1}{R_q} \sum_{k=1}^{n} P(k) \cdot \delta(k)$$

where $R_q$ is the number of relevant documents for query $q$, $P(k)$ is the precision at position $k$, and $\delta(k)$ is an indicator function that equals 1 if the document at position $k$ is relevant, otherwise 0. MAP is the mean of AP over all queries.

The **Pair Classification** task determines whether two sentences are semantically equivalent. An example dataset is Ocnli, which focuses on classifying sentence pairs into categories such as entailment, contradiction, or neutral. The evaluation metrics used are cosine similarity-based accuracy and Pearson correlation. Cosine similarity between embedding vectors $\mathbf{u}$ and $\mathbf{v}$ is defined as:

$$\cos(\mathbf{u}, \mathbf{v}) = \frac{\mathbf{u} \cdot \mathbf{v}}{\|\mathbf{u}\| \|\mathbf{v}\|}$$

Cosine similarity-based accuracy measures the alignment between predicted similarity and semantic equivalence, while Pearson correlation evaluates the linear relationship between cosine similarity scores and human-labeled ground truth.

The **STS (Semantic Textual Similarity)** task evaluates the degree of semantic similarity between pairs of sentences by comparing their embeddings. For example, the ATEC dataset assesses sentence similarity in financial question matching scenarios. The primary evaluation metric used is Pearson correlation, which quantifies the linear relationship between the predicted similarities and the ground truth labels. The Pearson correlation coefficient between predicted cosine similarities $\hat{y}$ and true similarities $y$ is computed as:

$$r = \frac{\sum_{i=1}^{n}(\hat{y}_i - \bar{\hat{y}})(y_i - \bar{y})}{\sqrt{\sum_{i=1}^{n}(\hat{y}_i - \bar{\hat{y}})^2} \sqrt{\sum_{i=1}^{n}(y_i - \bar{y})^2}}$$

The **Retrieval** task evaluates a model's ability to identify relevant documents in a large collection. For example, the MMarcoRetrieval dataset involves retrieving relevant documents for search queries. The primary evaluation metric is Recall at 1000, defined as:

$$\text{Recall@1000} = \frac{N_r}{N_t}$$

where $N_r$ is the number of relevant documents retrieved within the top 1000 results, and $N_t$ is the total number of relevant documents.

A.5    DETAILS OF THE WORD EVALUATION: TASK DESCRIPTION AND METRICS

We consider a four-choice question in the word evaluation, where each question has four options. For each question, we extract the word embeddings corresponding to the target word $w_i$ using different methods (ReBA, Echo, Classical) (see Table 1). After obtaining the four word embeddings, we calculate the pairwise Euclidean distances between the four vectors and select the word with the largest sum of distances to the other three vectors as the answer. The Euclidean distance between two vectors $\mathbf{u}$ and $\mathbf{v}$ is:

$$d(\mathbf{u}, \mathbf{v}) = \|\mathbf{u} - \mathbf{v}\|$$

In addition, we also consider the cosine distance, which measures the similarity between vectors as: The results using cosine distance are shown in figure 5.

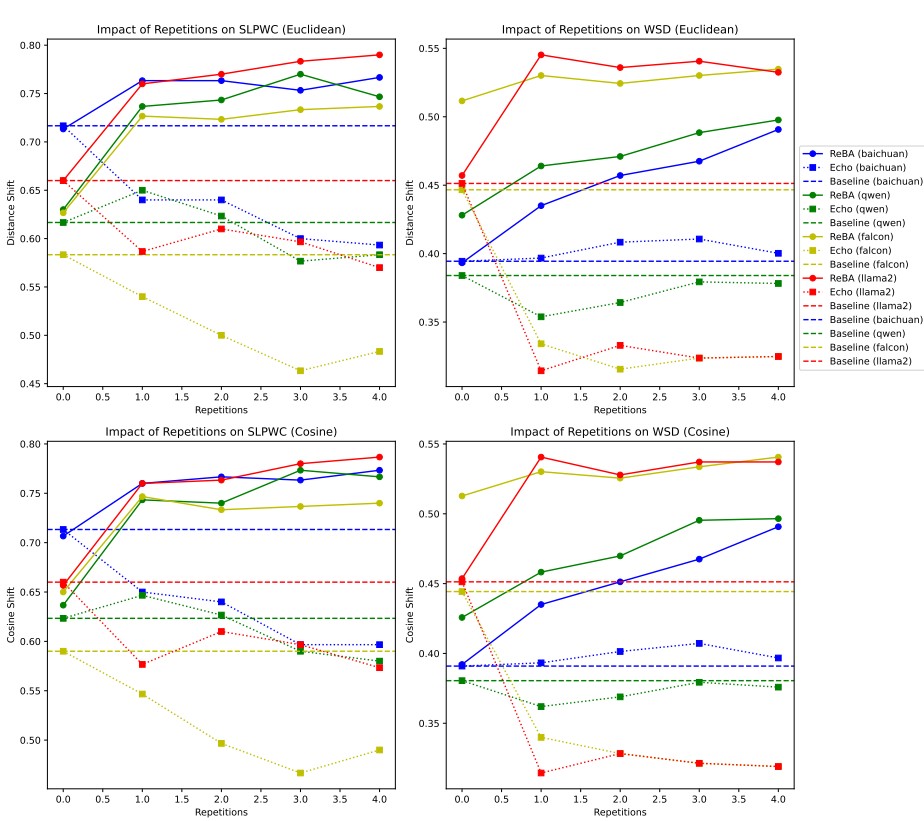

Figure 5: Performance on SLPWC and WSD tasks using Euclidean and Cosine distances to evaluate *word embeddings*, it shows that our results still hold under different distances,

