# OpenReview forum: "Retrieval Backward Attention without Additional Training: Enhance Embeddings of Large Language Models via Repetition"
_ICLR.cc/2026/Conference — ICLR 2026 Conference Withdrawn Submission_

### Official Review · Reviewer_g9zy · 2025-10-27

**Soundness:** 2
**Presentation:** 3
**Contribution:** 2
**Rating:** 2
**Confidence:** 4

**Summary:**

The paper proposes a method for improving word and sentence embeddings from causal LMs. To do this, the sentence to be embedded is repeated following prior work. Then, a backward attention method is developed: for a given word w_i, a new attention vector is created by taking the maximum value from any head/layer for each future word back to w_i. The new embedding for w_i is a sum of output embeddings of words k>i according to the weights in its new attention vector.

Experiments show that this method improves over alternative techniques for sentence embedding including Echo embeddings and small bidirectional models. For word embedding tasks, the proposed method improves over other causal methods but falls short of bidirectional encodings.

**Strengths:**

Strengths of this work include the improved performance of the proposed method.

The proposed backwards attention method is original as far as I know.

**Weaknesses:**

The improvements over echo sentence embeddings is often small. There is no report of statistical significance of these results or variance across multiple runs.

While the proposed method improves over other causal word embeddings, it still falls well short of even small bidirectional models.

The proposed backward attention is somewhat arbitrary. The weight between words w_i and w_k is determined by the maximum attention from w_k to w_i in any head/layer of w_k’s encoding, but this weight is then applied to the w_k output representation instead of the corresponding head/layer’s representation of w_k. This seems to work in practice, but it’s hard to justify when there are 1000+ layer/head weights in e.g. Llama 2 which represent queries (and keys) with unique transformations.

An analysis of the weights in the new attention matrix would help clarify what information is incorporated into the proposed embeddings which is perhaps missing from echo embeddings.

**Questions:**

Is the new attention vector normalized in any way? Is it a probability distribution?

What are typical values in the new attention vectors? I can imagine that across many head/layers for w_k there would be at least 1 which has a high value for w_i, even if this is due to noise.

---

### Official Review · Reviewer_A8gm · 2025-10-28

**Soundness:** 2
**Presentation:** 1
**Contribution:** 2
**Rating:** 2
**Confidence:** 5

**Summary:**

The paper proposes ReBA (Retrieval Backward Attention), which improves text embeddings from autoregressive LLMs by repeating input text and using extracted attention matrices to apply backward attention for updating token embeddings. Tested on C-MTEB with GPT-2 and LLaMA-2, ReBA achieves +7.3 points over Echo embeddings on GPT-2 and +0.7 on LLaMA-2, with particularly strong gains on clustering (+9.9) and retrieval (+4.2) tasks for GPT-2. The method is training-free, includes ablations on pooling strategies and repetition count, and demonstrates that backward attention provides meaningful improvements over simple text repetition, especially for smaller models

**Strengths:**

* It shows meaningful gains over Echo embeddings (+7.3 points), with notable improvements on clustering and retrieval tasks.

* It compares directly to Echo embeddings, allowing readers to assess the contribution of backward attention beyond simple repetition.

* It provides ablations on pooling strategies and repetition counts, offering some insights into implementation decisions.

* Authors propose an approach that doesn't require fine-tuning, which could be useful for practitioners with limited compute resources.

**Weaknesses:**

* Preliminaries is unnecessarily long and simple by ICLR standarts, and it lacks a related work section where methodology should be discussed with other training-free or lightweight embedding improvement techniques.

* Methodology should be primarily evaluted on english MTEB to compare with other methodologies since paper only compares with echo embeddings. Another problem is that mean pooling with BERT is already better than many methods in C-MTEB whereas this is not the case in English MTEB as shown in the echo embedding paper. Therefore, methodology might not translate to other languages or have marginal gains over echo embeddings. Moreover, it needs to compare to some trained baseline so we could see the benefit of not training such as LLM2Vec comparison in Echo Embedings.

* Tested models are quite limited, neither GPT-2 nor LLama2 are used in modern embedding models. For example , Mistral is a popular choice where it has very strong zero-shot performance.

* Ablation on only backward attention without repetition is missing.

**Questions:**

N/A

---

### Official Review · Reviewer_eKAT · 2025-11-01

**Soundness:** 2
**Presentation:** 2
**Contribution:** 2
**Rating:** 4
**Confidence:** 3

**Summary:**

The paper proposes ReBA, a method for extracting effective sentence and, more importantly, word embeddings from autoregressive language models. To achieve this, ReBA first constructs an attention matrix, and for each input token, recalculates its output embedding as an attention-weighted sum of all token embeddings in the sequence. The final sentence and word embeddings are then obtained in the same way as in classical baselines, but using these recalculated token embeddings. Experiments on C-MTEB and two word sense disambiguation datasets demonstrate the effectiveness of ReBA.

**Strengths:**

(1) This paper tackles the important problem of simultaneously extracting word and sentence embeddings from autoregressive language models.

(2) Experimental results at both the sentence and word levels demonstrate that the proposed methods outperform classical and echo-embedding baselines.

**Weaknesses:**

(1) Incomplete set of baselines. The paper lacks comparison with methods that extract sentence embeddings using last-token pooling with prompts applied (i.e., PromptEOL [1] and its follow-up works). Without this comparision, it is hard to fully verify the effectiveness of ReBA.

(2) Use of outdated models. The models used in the paper (GPT-2, LLaMA-2) are quite old. It is unclear whether the findings in the paper generalize to new models like Qwen3.

(3) Limited performance gains. Although ReBA requires no training, its performance with a 7B LLaMA model is only marginally higher than embeddings from BERT-base (e.g., 0.4801 vs. 0.4658 on C-MTEB) and even worse than BERT on word-level tasks. This limits the practical usefulness of the proposed approach.

[1] Jiang et al. Scaling Sentence Embeddings with Large Language Models. EMNLP 2024 Findings.

**Questions:**

(1) Some citation format is incorrect, e.g., "(Jelassi et al., 2024) found that" should be "Jelassi et al., 2024 found that".

(2) Since ReBA contains no specific design for Chinese, is there any reason for selecting C-MTEB instead of rather than the English MTEB as the evaluation benchmark?

(3) In section 3.2, you mention that ReBA does not use prompt. However, isn't adding words "Repated once:" also some form of prompt?

(4) In Line 217, you state that "due to causal masking, w5 cannot access information
from the subsequent token w6 and only incorporates context from the preceding w4,". However, given a sequence {w1, w2, w3, w4, w5, w6}, where {w1, w2, w3} form the original input, w5 can already access information from the entire original input. In this case, does the absence of information from w6 actually matter?

(5) It is unclear to me how use the maximum value observed across all heads and layers as the final extracted attention weight can enhance interpretability. Could you explain this a bit more?

---

### Official Review · Reviewer_nU6a · 2025-11-01

**Soundness:** 2
**Presentation:** 2
**Contribution:** 2
**Rating:** 4
**Confidence:** 4

**Summary:**

This paper introduces ReBA, a method for enhancing embeddings from decoder-only LLMs without any additional training. The key idea is to combine input repetition with a backward attention mechanism that allows tokens to integrate information from subsequent context. The authors demonstrate improvements on both sentence- and word-level tasks, particularly on the Chinese MTEB benchmark and polysemous word disambiguation datasets. ReBA is conceptually simple and computationally efficient, relying only on existing model attention matrices.

**Strengths:**

1. This paper addresses a real limitation of decoder-only models: their inability to incorporate future context when generating embeddings. The problem is well-motivated, and the proposed approach fits naturally within current trends toward training-free post-processing of LLM representations.
2. ReBA is algorithmically straightforward, requiring only access to the model’s attention maps and repetition of inputs.

**Weaknesses:**

1. For larger and stronger LLMs such as LLaMA-2, the performance gain from ReBA appears marginal compared to a simple echo-based baseline. This suggests that powerful models may already encode much of the contextual redundancy ReBA tries to add.
2. This paper does not include results on the English MTEB benchmark suite. Evaluating ReBA on that dataset would provide stronger evidence of its generality.

**Questions:**

See above.

---

### Note · Authors · 2026-01-17

I have read and agree with the venue's withdrawal policy on behalf of myself and my co-authors.